# Temporal selectivity declines in the aging human auditory cortex

**Julia Erb\*, Lea-Maria Schmitt, Jonas Obleser**

Department of Psychology, University of Lübeck, Lübeck, Germany

**Abstract** Current models successfully describe the auditory cortical response to natural sounds with a set of spectro-temporal features. However, these models have hardly been linked to the ill-understood neurobiological changes that occur in the aging auditory cortex. Modelling the hemodynamic response to a rich natural sound mixture in N = 64 listeners of varying age, we here show that in older listeners' auditory cortex, the key feature of temporal rate is represented with a markedly broader tuning. This loss of temporal selectivity is most prominent in primary auditory cortex and planum temporale, with no such changes in adjacent auditory or other brain areas. Amongst older listeners, we observe a direct relationship between chronological age and temporal-rate tuning, unconfounded by auditory acuity or model goodness of fit. In line with senescent neural dedifferentiation more generally, our results highlight decreased selectivity to temporal information as a hallmark of the aging auditory cortex.

## Introduction

Age-related hearing loss is a frequent cause for a decline in speech comprehension, particularly in complex acoustic environments. As the temporal envelope of speech is crucial for speech comprehension (*Shannon et al., 1995*), the ability to accurately encode temporal features in the auditory system is pivotal for successful perception of speech and conspecific vocalizations (*Peelle and Wingfield, 2016*). Age-related deficits have been observed in different temporal psychoacoustic tasks (e.g. gap detection; *Snell et al., 2002*). Those psychoacoustic studies suggest that the precision of processing of temporal cues declines with age (e.g. *Gordon-Salant et al., 2006*).

While age-related hearing loss has been attributed to a variety of peripheral declines in auditory coding (e.g. *Gates and Mills, 2005*), fewer studies have looked at more central levels of auditory processing. Yet, declines in temporal processing are hypothesized to be not exclusively peripheral in origin, but more central than at least the mid-brain (*Recanzone, 2018*). In the aged macaque auditory cortex, neurons become more broadly tuned to temporal modulations and temporal fidelity of cortical responses decreases (*Ng and Recanzone, 2018*). Topographically, a neural dedifferentiation is suggested to take place in the older brain: In young macaques, the neuronal response to the inter-stimulus interval in primary field A1 and caudolateral belt area (CL) differs in that A1 neurons have shorter response latencies. In aged animals, however, no difference was observed between A1 and CL such that aged A1 neurons had an equivalent response latency to CL neurons (*Ng and Recanzone, 2018*). Further, in aged neurons, a shift in neural coding strategy from less temporal coding towards more rate coding of temporal modulations was evident (*Overton and Recanzone, 2016*; for a review of both coding strategies see *Joris et al., 2004*). For those neurons that still adhered to a temporal coding strategy, the temporal fidelity decreased, although the absolute number of neurons responsive to temporal modulations was unaffected by age (*Overton and Recanzone, 2016*).

At higher processing levels, only a number of relatively unspecific changes have been reported in the older human listener (for review see *Peelle and Wingfield, 2016*). For example, the frontal and cingulo-opercular cortical response is elevated during a challenging speech comprehension or gap

**\*For correspondence:**
julia.erb@uni-luebeck.de

**Competing interest:** See
page 17

**Reviewing editor:** Ingrid S
Johnsrude, University of Western
Ontario, Canada

**eLife digest** It can often be difficult for an older person to understand what someone is saying, particularly in noisy environments. Exactly how and why this age-related change occurs is not clear, but it is thought that older individuals may become less able to tune in to certain features of sound.

Newer tools are making it easier to study age-related changes in hearing in the brain. For example, functional magnetic resonance imaging (fMRI) can allow scientists to 'see' and measure how certain parts of the brain react to different features of sound. Using fMRI data, researchers can compare how younger and older people process speech. They can also track how speech processing in the brain changes with age.

Now, Erb et al. show that older individuals have a harder time tuning into the rhythm of speech. In the experiments, 64 people between the ages of 18 to 78 were asked to listen to speech in a noisy setting while they underwent fMRI. The researchers then tested a computer model using the data. In the older individuals, the brain's tuning to the timing or rhythm of speech was broader, while the younger participants were more able to finely tune into this feature of sound. The older a person was the less able their brain was to distinguish rhythms in speech, likely making it harder to understand what had been said.

This hearing change likely occurs because brain cells become less specialised overtime, which can contribute to many kinds of age-related cognitive decline. This new information about why understanding speech becomes more difficult with age may help scientists develop better hearing aids that are individualised to a person's specific needs.

detection task (*Erb and Obleser, 2013*; *Vaden et al., 2015*; *Vaden et al., 2020*). This has been interpreted as compensatory mechanism in response to loss of sensory acuity. Aging has been hypothesized to lead to a neural dedifferentiation in sensory cortices (e.g., *Park et al., 2004*). A more recent unspecific observation is that with age, decoding accuracy of stimulus conditions from auditory cortical fMRI responses declines (*Lalwani et al., 2019*; but see discussion for contrasting findings in MEG speech tracking, for example *Presacco et al., 2016*). This finding in turn has been linked to an imbalance of excitation and inhibition in older adults, more specifically a reduction of GABA levels in auditory cortex (*Lalwani et al., 2019*). Accumulating evidence supports the theory of an age-related loss of inhibition in sensory cortices (*Caspary et al., 2008*).

A recent model of cortical processing (*Chi et al., 2005*) has led to substantial progress in our understanding of how natural sounds become represented in the auditory cortex. A series of studies have shown that cortical processing of sounds is optimized to represent the spectro-temporal modulations which are typically present in conspecific vocalizations such as speech or animal calls (*Santoro et al., 2014*; *Hullett et al., 2016*; *Santoro et al., 2017*; *Erb et al., 2019a*). However, those studies have not looked at the pressing case of the aged human cortex. It remains unknown whether — and if so, which — general response properties are altered in aged auditory cortex.

The present study applies an fMRI encoding and decoding approach to compare auditory cortical responses to natural sounds in young and older humans. We focus on the age-group comparison at the level of representation of fundamental acoustic features (spectro-temporal modulations). Results show that while the large-scale topographic organization of acoustic features is preserved in the older auditory cortex, age groups differ in tuning to temporal modulations: Human aging thus appears to be accompanied by an anatomically and functionally specific broadening of temporal-rate tuning in auditory cortex.

## Results

fMRI responses to sounds from a total of N = 64 younger and older listeners were modelled using both model-based encoding (*Santoro et al., 2014*) and decoding (*Santoro et al., 2017*; *Figure 1*). In brief, sounds were decomposed into acoustic features, that is, the frequency-dependent spectro-temporal modulation content (*Chi et al., 2005*). Analyses were restricted to voxels in an anatomic mask of the auditory cortex. In a fourfold cross-validation procedure, we used ridge regression to derive single- and multi-voxel MTFs.

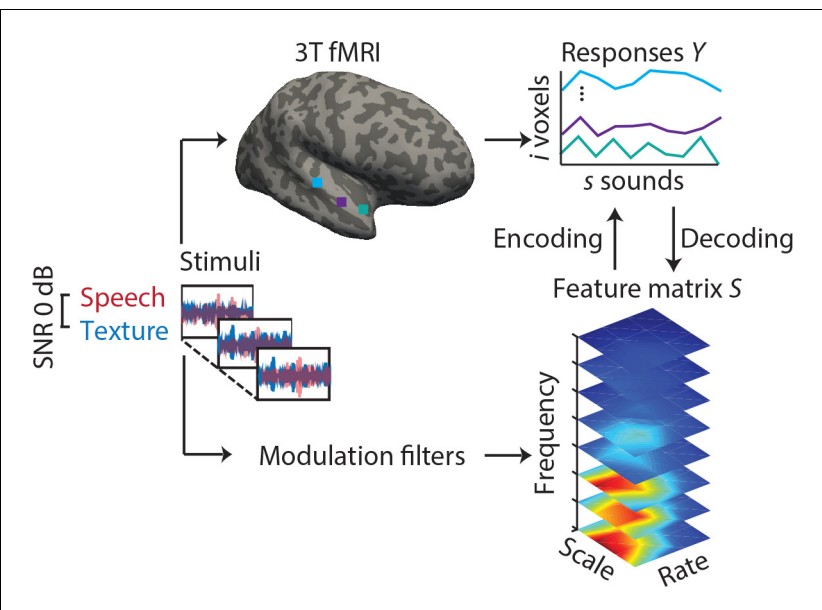

**Figure 1.** Experimental design and analysis pipeline. Participants listened to a story embedded in sound textures at a signal-to-noise ratio(SNR) of 0 dB while we acquired 3T-fMRI data. For audiograms of the participants see *Figure 1—figure supplement 1*). The sounds were decomposed into their modulation spectrum (*Chi et al., 2005*; for modulation spectra of the sounds see *Figure 1—figure supplement 2*). In a first *univariate encoding* approach, we calculated a modulation transfer function (MTF) for each individual voxel. In a second *multivariate decoding* approach, linear decoders were trained on the response patterns in auditory cortex for each feature of the modulation representation. Predictions were then tested on a left-out testing data set.

The online version of this article includes the following figure supplement(s) for figure 1:

**Figure supplement 1.** Pure-tone audiometry and behavioural results.
**Figure supplement 2.** Modulation spectra of the stimuli.

## Encoding results

We compared three models of hemodynamic responses to sound. The models respectively describe the fMRI responses to the (1) speech stream (foreground), (2) textures (background), or (3) the mixture of both streams (for modulation spectra of the different streams see *Figure 1—figure supplement 2*).

### Sound identification accuracies

In a first encoding analysis, we estimated a modulation transfer function (MTF) for each voxel based on a subset of fMRI data (training) for each model. We then assessed the ability of these models to accurately predict the fMRI responses to sounds of a new, independent data set (testing data). We quantified prediction accuracy by means of a sound identification analysis (see Materials and methods).

Sound identification accuracy was higher than chance level (0.5) for all models and age groups (*Figure 2*). The sound mixture model had the highest sound identification accuracies (*Figure 2a*) and thus described most accurately the fMRI responses to sounds. All subsequent encoding and decoding analyses were based on this mixture model. Young participants had significantly higher identification accuracies for mixture than old participants (*Figure 2a*). There was no significant correlation between chronological age and sound identification accuracy for mixture within the young or older group (*Figure 2b*).

### Topographical best-feature maps

Best feature maps were obtained from univariate encoding models by marginalizing the single-voxel MTFs for the dimension of interest. In particular, tonotopic maps were obtained by averaging over the rate and scale dimension and assigning the frequency with the maximal response to a given

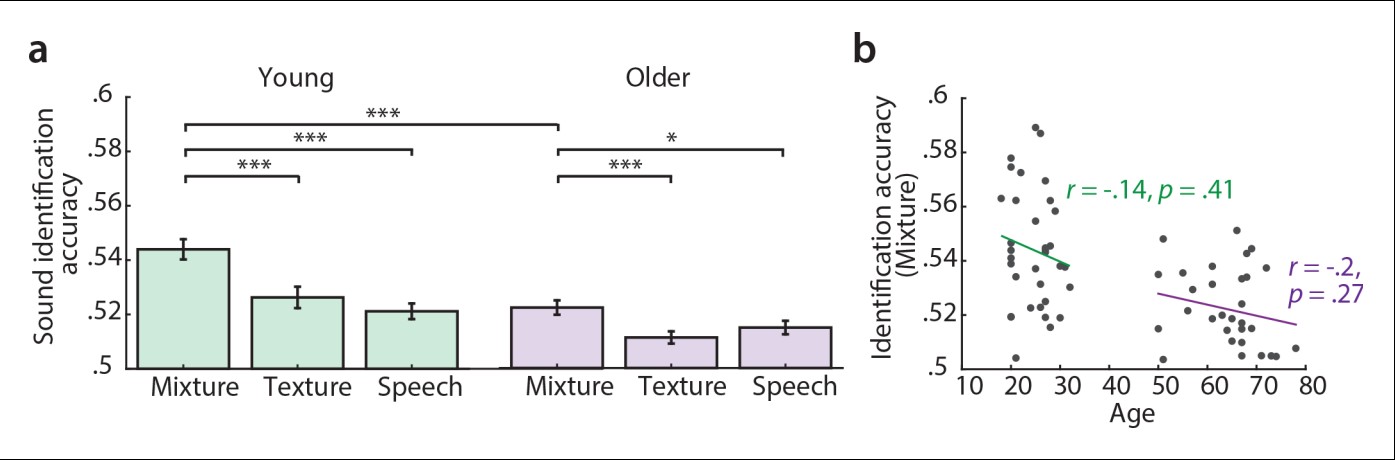

**Figure 2.** Sound identification accuracies for the different encoding models. (a) Accuracies for sound mixture, texture (background stream) and speech (foreground stream), for young (left, green) and older participants (right, violet). Bars indicate the mean (± standard error of the mean, SEM) of all participants per age group. Accuracies are normalized between 0 and 1; zero denotes that the predicted activity pattern for a given stimulus was least similar to the measured one among all test stimuli; one denotes correct identification; chance level is 0.5. *p<0.05, **p<0.01, ***p<0.005, exact permutation test. (b) Identification accuracies for sound mixture were uncorrelated with chronological age within each age group.

voxel. Tonotopic maps showed the typical mirror-symmetric frequency gradients along Heschl's gyrus in both young and aged auditory cortex, irrespective of age (*Figure 3*). Tonotopic gradients were most obvious in the young group (*Figure 3a*, upper panel). Consistent with previous human fMRI data (*Moerel et al., 2012*), a low frequency region was observed in the central region of Heschl's gyrus (HG), presumably marking the boundary between the human homologue of primary

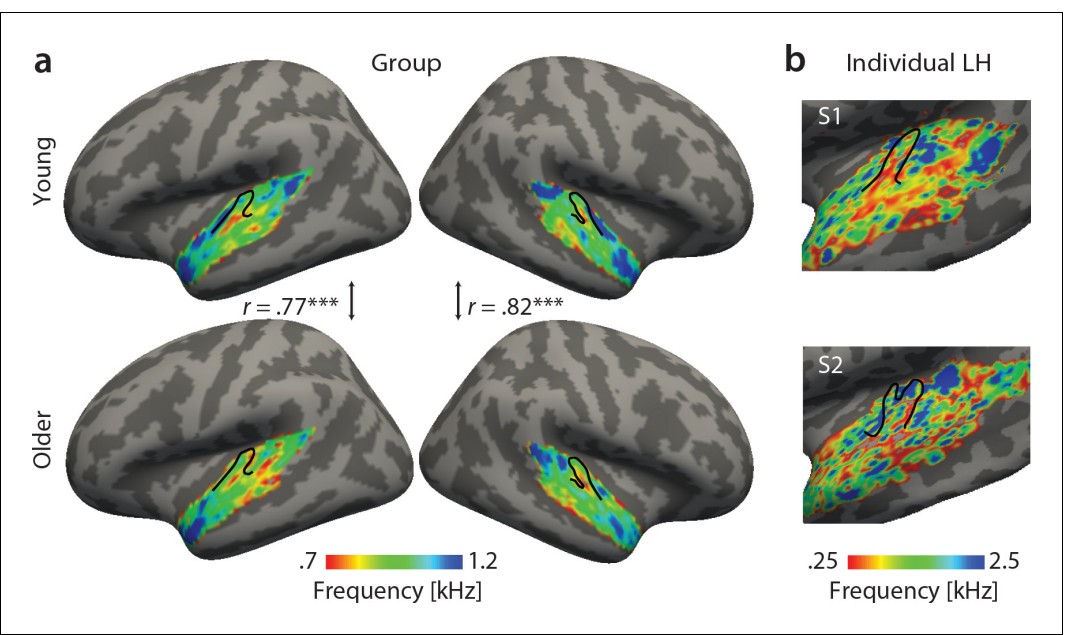

**Figure 3.** Tonotopic maps. Best frequency maps were derived by marginalizing modulation transfer functions (MTFs) for frequency and assigning the feature with the maximal response to a given voxel. (a) Group maps were obtained as median across subjects and are shown on FreeSurfer's *fsaverage5* template. Tonotopic maps in the young (upper panel) and older group (bottom panel) are correlated in both the right and left hemisphere. (b) Exemplary individual tonotopic maps for the left hemisphere (LH) of a young (upper panel) and older participant (bottom panel) are displayed on individual surfaces. Black outlines indicate Heschl's gyrus. ***p<0.005.

auditory fields A1 and R (*Figure 3a*, red-yellow). This low frequency region was surrounded antero-medially and posteriorly by high frequency regions (*Figure 3a*, green-blue). The antero-medial high frequency areas clustered on the planum polare (PP). The posterior regions preferring high frequencies covered planum temporale (PT). We did not find significant age-group differences using two-sample voxel-wise *t*-tests (corrected for multiple comparisons). In fact, tonotopic group maps were correlated, indicating that topography for best frequency was preserved in older adults (*Figure 3a*).

Topographical best-feature maps for rate and scale were more complex. Similarly, we did not find significant voxel-wise age-group differences using two-sample *t*-tests (corrected for multiple comparisons). On the contrary, both for temporal and spectral modulations, group maps were correlated between age groups (*Figure 4*). For best temporal modulation maps, we observed a medial-to-lateral gradient for increasing temporal rate (*Figure 4a*). For best spectral modulation maps, the locus of highest spectral resolution coincided with the low frequency region on Heschl's gyrus (*Figure 3a*, *Figure 4b*).

### Decoding results
#### Feature reconstruction from auditory cortex
Cortical sensitivity to acoustic features was quantified using model-based decoding. Decoders were trained on a subset of the data (training) for each acoustic feature separately (*Figure 1*). Reconstruction accuracies were obtained as Pearson's *r* between predicted and actual acoustic features in the testing data. Results were compared across age groups using exact permutation tests.

Our main hypothesis was that cortical sensitivity is highest for slow temporal modulations, based on previous observations of human (but not monkey) auditory cortex being most sensitive to the modulations present speech (*Santoro et al., 2017*; *Erb et al., 2019a*). Decoding yielded highest accuracies at frequencies of 230–580 Hz (mean *r* = 0.47) and spectral scales of 0.25 cyc/oct (mean *r* = 0.47), irrespective of age (*Figure 5a,b*), indicating that brain responses followed the frequency

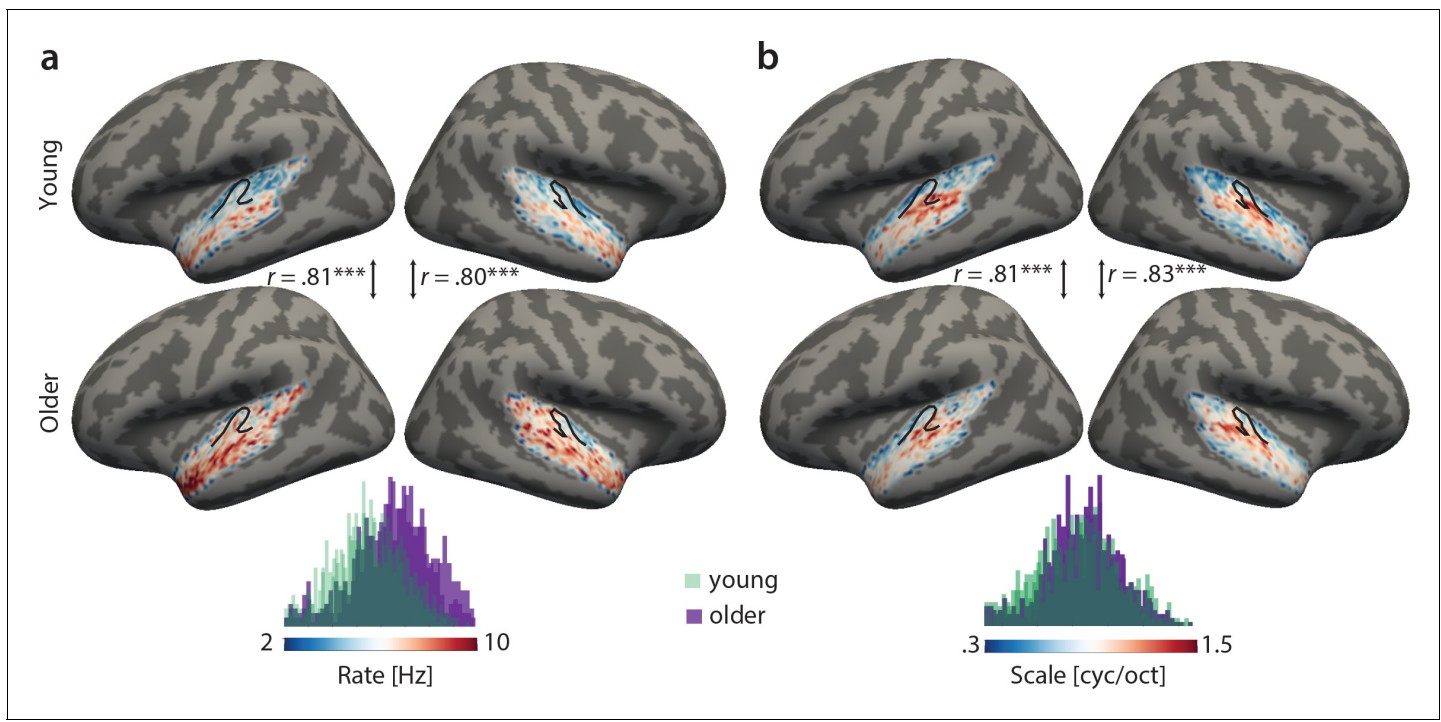

**Figure 4.** Group best-feature maps for temporal rate (a) and spectral scale (b). Group maps were obtained as median across subjects and shown on FreeSurfer's *fsaverage5* template. Black outlines indicate Heschl's gyrus. Topographic organization in the young (upper panel) and older group (bottom panel) is significantly correlated (Pearson's correlation coefficient) for best temporal rate (a) and best spectral scale (b). Histograms of best feature values are shown for the young (green) and older group (violet) above the colour bars. Note the shift of the distribution in older relative to young adults for temporal rate (a) but not spectral scale (b), indicating that older adults' voxel-wise preferred temporal rates are shifted towards higher modulation rates. ***p<0.005.

and spectral modulation content of the stimuli (*Figure 1—figure supplement 2a*). Conversely, for temporal modulations, reconstruction accuracies peaked at rates of 4–8 Hz (mean *r* = 0.4) in both age groups (*Figure 5a,b*). Those peaks were not present in the stimuli (*Figure 1—figure supplement 2a*) and imply preferred processing of temporal rates in the speech-relevant range (*Poeppel and Assaneo, 2020*), corroborating our hypothesis (see also discussion).

### Selectivity index

A key observation was the age difference in selectivity of tuning to temporal modulations (as quantified by the selectivity index *SI*, see Materials and methods). Selectivity of temporal rate coding was higher in young than in older participants (mean *SI* age difference = 0.01, p=0.004 [permutation test], Cohen's *d* = 0.68, *Figure 5d*, left). Note that removing the extreme case in the young group (outliers were defined as exceeding the grand average by ±2 SD) for rate *SI* does not alter results qualitatively: It reduces the mean age-group difference to a still significant *SI* difference = 0.009, p=0.026. Removing the two extreme cases in the older group for scale *SI* increases the mean age-group difference to a significant *SI* difference = 0.037, p=0.024. A lower selectivity index is consistent with a broadening of tuning in the aged auditory cortex.

Careful artefact removal from fMRI data is particularly important for age comparisons. Therefore, we re-ran the decoding analysis on ICA-cleaned data (using AROMA, *Pruim et al., 2015*).

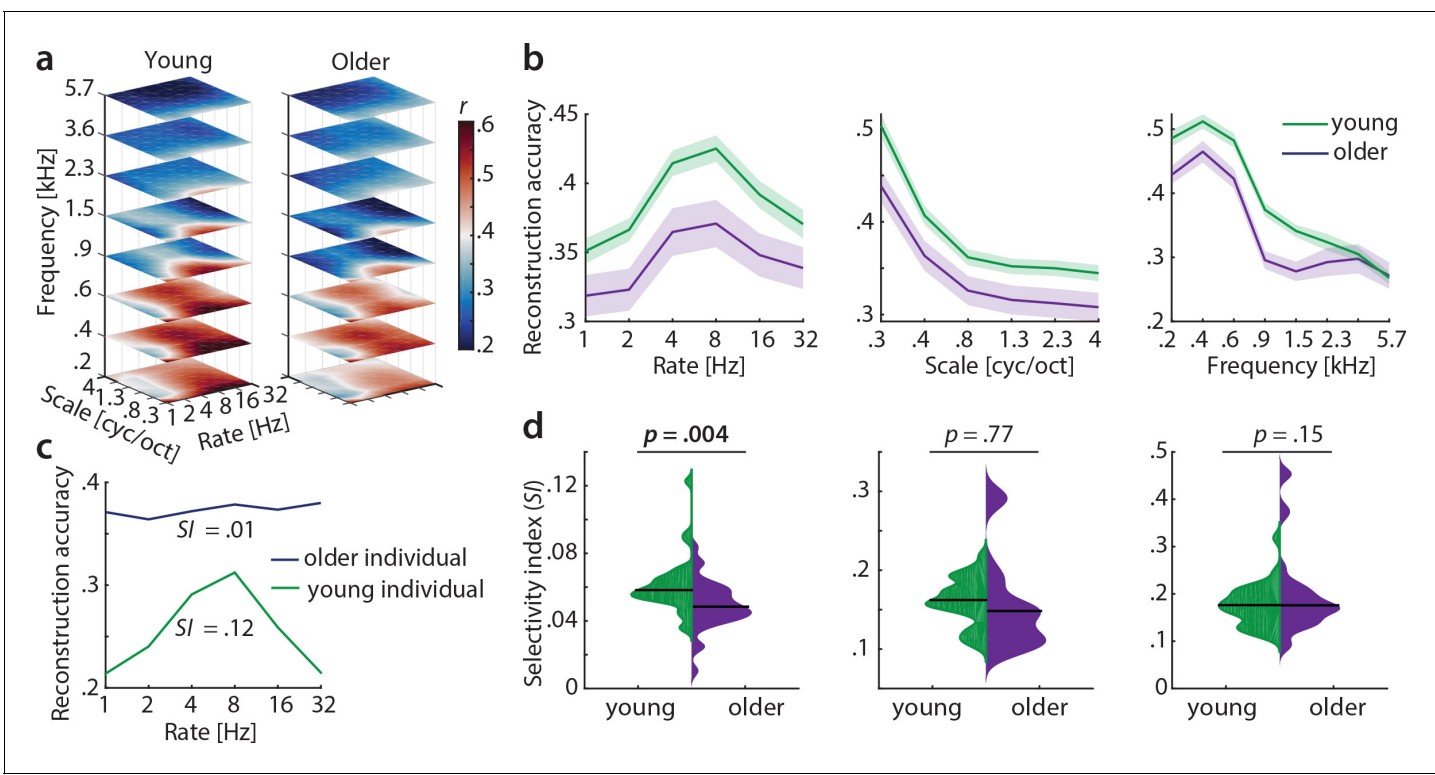

**Figure 5.** Decoding from auditory cortex. (a) Auditory cortical modulation transfer functions (MTFs) for the young and older group quantified by reconstruction accuracies (Pearson's *r* between predicted an actual acoustic features). (b) Mean ± SEM of the MTFs' marginal profiles for rate, scale and frequency for young (green) and older (violet) participants separately. (c) We quantified tuning selectivity by the selectivity index (*SI*; see *Equation 7*). For illustration purposes, we show the marginal profiles for rate in a young (green) and older individual (violet) with extreme *SI* values. (d) *SI* was compared for each acoustic dimension (rate, scale, frequency) between age groups using an exact permutation test. Black line indicates the median *SI*. Black line indicates the median *SI*. Note that when removing the extreme case in the young group (exceeding the grand average by ±2 SD) for rate *SI*, the mean age-group difference remains significant (p=0.026). Removing the two extreme cases in the older group for scale *SI* leads to a significant age-group difference (p=0.024). Note the different scaling of the plots. See *Figure 5—figure supplement 1* for decoding from auditory cortex with ICA (AROMA)-cleaned data.

The online version of this article includes the following figure supplement(s) for figure 5:

**Figure supplement 1.** Decoding from auditory cortex with ICA(AROMA)-cleaned data.

Irrespective of preprocessing without (*Figure 5*) or with ICA-cleaning (*Figure 5—figure supplement 1*), the key results remained unchanged, that is, reconstruction accuracies peaking at temporal rates of 4–8 Hz as well as selectivity for temporal rates being higher in young than in older participants.

Also, this key result of a broadened tuning to temporal rate was corroborated by an even simpler measure of selectivity: The variance in the reconstruction accuracy profiles across the six temporal rate bins (1–32 Hz, 'rate variance') was higher in the young than in the older participants (mean difference in variance = $3.7 \times 10^{-4}$, p<0.001 [permutation test], Cohen's $d$ = 0.93, *Figure 6* a, left).

In the older group, the rate variance was significantly correlated with the rate selectivity index and chronological age but was uncorrelated with hearing loss (*Figure 6*, b). Note however that the rate selectivity index (see above) in the older group was not significantly correlated with chronological age (Spearman's rho = $-0.2$, p=0.289), nor with hearing loss (Spearman's rho = 0.34, p=0.062).

In a control analysis, we ensured that these results cannot be explained by the overall slightly worse model fits in the older group as quantified by sound identification accuracy (as evident in *Figure 2*): In a multiple regression predicting older participants' variance in rate from chronological age with covariates PTA and sound identification accuracy, neither PTA ($t(26)$ = 0.73, p=0.473, permutation test) nor sound identification accuracy ($t(26)$ = 1.02, p=0.312) were significant, while age was significant ($t(26)$ = $-2.8$, p=0.003; *adjusted $R^2$* = 0.39). Age remained significant when excluding two multivariate outliers identified based on Cook's distance ($t(24)$ = $-2.25$, p=0.021).

### Feature reconstruction from auditory subregions

To test the differential sensitivity of auditory cortical subregions to acoustic features, we re-ran the decoding analyses in the following regions of interest (ROIs) derived from the FreeSurfer atlas (*Destrieux et al., 2010*): Heschl's gyrus and sulcus (HG/HS), planum polare (PP), planum temporale

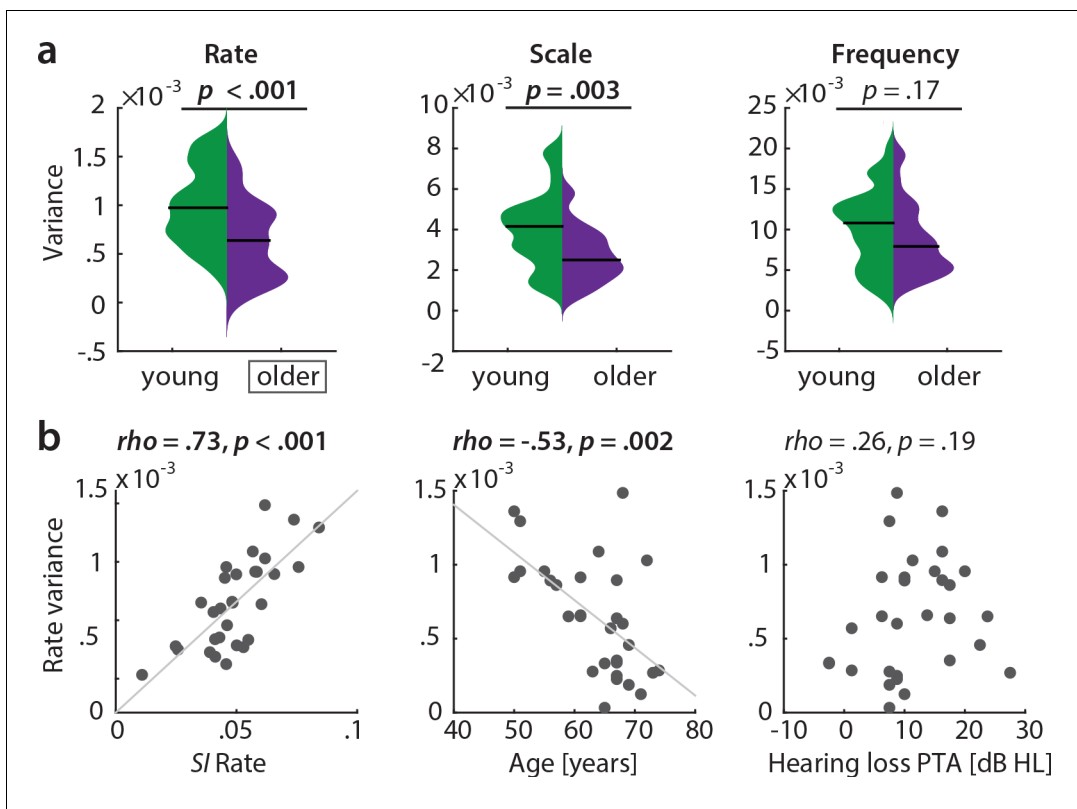

**Figure 6.** Variance along reconstruction accuracy profiles as alternative measure for cortical tuning selectivity. (**a**) Variance of the MTFs' marginal profiles compared between age groups. *P*-values are based on an exact permutation test. (**b**) Amongst the older group, the variance of the rate profiles ('rate variance') correlates with the *SI* for rate (left panel) and chronological age (middle panel), but not with hearing loss (right panel; Spearman's correlation coefficient). MTF: modulation transfer function, *SI*: selectivity index, PTA: pure tone average.

(PT), superior temporal gyrus (STG) and superior temporal sulcus (STS, *Figure 7a - f*)). As control regions, we chose primary visual cortex (V1, calcarine sulcus) as another primary sensory area, and two higher-level areas, namely the middle frontal gyrus (MFG) and superior parietal gyrus (SPG; *Figure 7a, g-i*).

A decrease in the rate selectivity index with age was confirmed for HG/HS (mean *SI* difference = 0.012, p=0.009 [permutation test]) and PT (mean *SI* difference = 0.016, p=0.005 [permutation test]; see *Figure 7b,c*). Removing the extreme cases in the young group exceeding the grand average by ±2 SD reduces the mean *SI* age difference to 0.012 (p=0.008) in HG/HS, and to 0.012 (p=0.04) in PT. The alternative measure of selectivity, variance of the rate profiles, was also significantly higher in young than older adults in HG/HS (variance difference = $4 \times 10^{-4}$, p=0.0004) and PT (variance difference = $5 \times 10^{-4}$, p=0.0003).

Note that reconstruction accuracies in all ROIs but the control regions (V1, SPG, MFG) were significant (permutation tests; for the z-scored reconstruction accuracies in an exemplary region HG/HS see *Figure 7—figure supplement 1*).

In contrast, there were no age differences in temporal selectivity in PP, STG, STS or the control regions V1, SPG and MFG (*Figure 7d* – i).

Qualitatively, this observation was confirmed by a control analysis, where we derived topographic maps of the rate selectivity index from the encoding models. To this end, we marginalized MTFs for rate and calculated the *SI* for a given voxel. The topographic distribution confirmed highest voxel-wise *SI* for rate in area PT of the younger group (*Figure 7—figure supplement 2*).

These results indicate that age-related changes in temporal selectivity selectively occur in primary auditory cortex and the planum temporale, but not surrounding belt areas. For spectral scale, we also found a decreased *SI* in older adults in HG/HS (*SI* difference = 0.012, p=0.008) and PT (*SI* difference = 0.016, p=0.021), but none of the other ROIs (*Figure 7—figure supplement 3*). For frequency, we did not find any age differences in selectivity index at all (*Figure 7—figure supplement 3*).

## Discussion

This cross-sectional approach used encoding and decoding models of hemodynamic responses to compare the spectro-temporal fidelity in auditory cortex of younger and older listeners. We pursued the long-standing but largely understudied question whether temporal and/or spectral modulations, which are key acoustic dimensions of behaviourally relevant signals such as speech and music, are represented in a less differentiated fashion in the aging auditory cortex.

We find the large-scale topographic organization of acoustic features entirely preserved in the aged auditory cortex, and best feature maps for frequency, scale and rate accordingly were correlated very highly between age groups.

However, age-related differences are evident in the cortical sensitivity to mainly temporal modulations as quantified by the decoding analysis. Decoding of temporal rate is most accurate at slow rates of 4–8 Hz irrespective of age, but the tuning to temporal rate is significantly sharper in young than in older participants. Also, amongst older adults, broadening of rate tuning correlates with chronological age. Importantly, the age-related rate selectivity difference was driven by primary auditory (HG) and adjacent area PT, while secondary auditory areas PP, STG, and STS did not show age differences.

### Preserved topographic organization in aging auditory cortex

Using natural stimuli, we revealed topographic maps of acoustic feature preference for frequency, spectral and temporal modulations. Thus, we show that topographic maps of feature preference can be observed under approximately natural hearing situations. Tonotopic maps exhibited the well-established mirror-symmetric high-low-high frequency gradients across Heschl's gyrus and adjacent areas (*Figure 3*) consistent with abundant evidence in human (*Formisano et al., 2003*; *Talavage et al., 2004*; *Humphries et al., 2010*; *Woods et al., 2010*; *Da Costa et al., 2011*; *Striem-Amit et al., 2011*; *Langers and van Dijk, 2012*; *Moerel et al., 2012*; *Moerel et al., 2014*) and non-human primates (*Merzenich and Brugge, 1973*; *Morel et al., 1993*; *Kosaki et al., 1997*; *Rauschecker et al., 1997*; *Bendor and Wang, 2008*; *Baumann et al., 2013*; *Joly et al., 2014*;

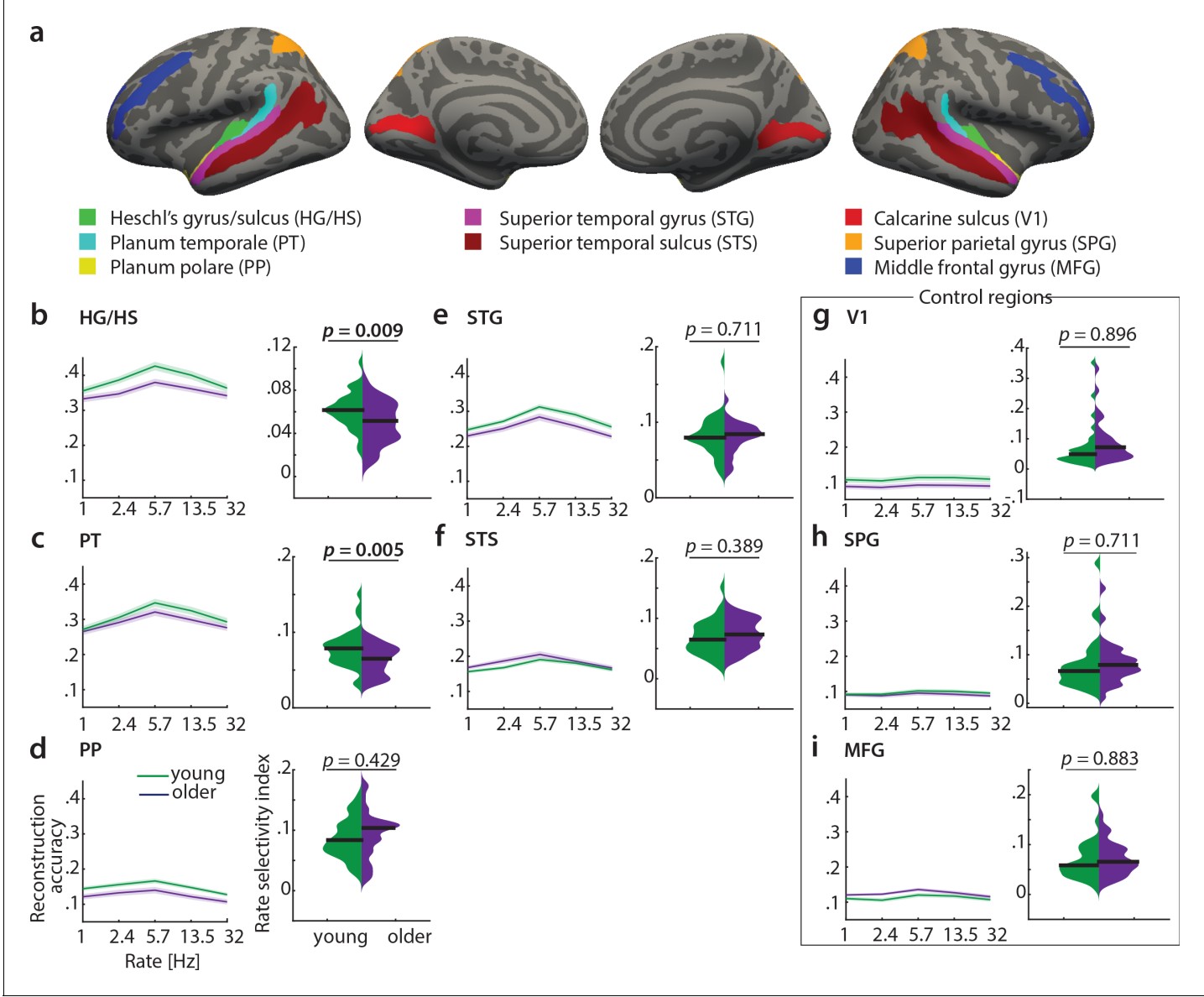

**Figure 7.** Decoding of temporal rate from regions of interest (ROIs). (a) ROIs derived from the FreeSurfer atlas (*Destrieux et al., 2010*) are displayed on FreeSurfer's *fsaverage* template. Mean (± SEM) reconstruction accuracy profiles (left) and selectivity index (right) per age group for temporal rate in (b) Heschl's gyrus and sulcus, (c) planum temporale, (d) planum polare, (e) superior temporal gyrus, (f) superior temporal sulcus and the control regions (g) calcarine sulcus, (h) superior parietal gyrus, (i) middle frontal gyrus. The selectivity index (*SI*) was compared per ROI between age groups using an exact permutation test. When removing the young-subject *SI* exceeding the grand average by ±2 SD, the *SI* age difference remains significant in HG/HS (p=0.008), and in PT (p=0.04). Consistently, our alternative measure of selectivity, variance across rate bins, was significantly higher in young than older adults in HG/HS (p=0.0004) and PT (p=0.0003). See *Figure 7—figure supplement 1* for the reconstruction accuracies z-scored with respect to the empirical null distribution in an exemplary region HG/HS. See *Figure 7—figure supplement 2* for group maps of the rate selectivity index obtained from encoding models. See *Figure 7—figure supplement 3* for decoding of spectral scale and frequency from ROIs.

The online version of this article includes the following figure supplement(s) for figure 7:

**Figure supplement 1.** Z-scored reconstruction accuracies from Heschl's gyrus/sulcus.

**Figure supplement 2.** Topographic maps for the rate selectivity index.

**Figure supplement 3.** Decoding of spectral scale and frequency in ROIs.

*Baumann et al., 2015*). Notably, tonotopic maps in younger versus older adults were highly correlated, indicating topographical stability of tonotopic organization across age groups.

The well-demonstrated existence of tonotopic maps notwithstanding, a topographic organization for temporal or spectral modulations in auditory cortex is less clear (*Schönwiesner and Zatorre, 2009*). Here, in both young and older adults we observed a medial-to-lateral low-to-high gradient for temporal modulations across the superior temporal plane (*Figure 4a*), while for spectral modulations, the locus with the highest resolution (~1.5 cyc/oct) was observed in lateral Heschl's gyrus (*Figure 4b*). A topographic representation for modulation rate has previously been observed in the primate auditory cortex (*Baumann et al., 2015*) where fast temporal acoustic information was preferably encoded in caudal auditory regions (*Camalier et al., 2012*; *Kusmierek and Rauschecker, 2014*) and slow temporal information in rostral areas R and RT (*Liang et al., 2002*; *Bendor and Wang, 2008*).

Whereas earlier human fMRI studies had failed to show a clear topography for modulation rate (*Giraud et al., 2000*; *Schönwiesner and Zatorre, 2009*; *Overath et al., 2012*; *Leaver and Rauschecker, 2016*), recent human fMRI (*Santoro et al., 2014*) and electrocorticography (ECoG) studies (*Hullett et al., 2016*) proposed the presence of a posterior-to-anterior high-to-low rate gradient in human auditory cortex. However, comparison of those studies is hampered by the differences in coverage and ranges of temporal modulations: While *Hullett et al., 2016* examined representation of slow temporal modulation rates present in speech (1–2 Hz) only along the STG with ECoG, *Santoro et al., 2014* analyzed rates of 1–27 Hz, whereas *Baumann et al., 2015* presented rates of up to 512 Hz with more comprehensive coverage using fMRI.

## Functional mapping from natural sounds is feasible using conventional MR field strength in aging populations

While previous studies had presented single natural sounds (*Erb et al., 2019a*), or even synthetic sounds such as AM stimuli (*Baumann et al., 2015*) or dynamic ripples (*Schönwiesner and Zatorre, 2009*), here, we presented a continuous stream of speech embedded in an acoustically rich background of sound textures. As an important foundation for the main conclusions of this manuscript, we here demonstrate the capacity to derive meaningful tonotopic maps from natural stimulus conditions, at conventional field strengths (3 T) and in special populations (older participants).

Although an inherent caveat of the use of natural stimulus conditions is the difficulty of measuring psychoacoustic performance, this experimental design has the advantage that it closely approximates natural listening conditions (*Hamilton and Huth, 2020*). As synthetic sounds lack both the behavioural relevance and the statistical structure of natural sounds, they activate auditory cortex differently than natural stimuli (*Theunissen et al., 2000*; *Bitterman et al., 2008*).

Methodologically, our encoding approach relies on the assumption that acoustic features that maximally activate single voxels are more accurately encoded. However, higher responses may not necessarily mean better encoding. The second, complementary decoding analysis incorporates the whole range of values rather than only the maximal values. The reconstruction accuracy explicitly quantifies the amount of information about a set of stimulus features that is available in a region of interest. Accurate reconstruction of acoustic features in test sounds indicates that these features are reproducibly mapped into distinct spatial patterns. As data from individual voxels are jointly modelled, this combined analysis of signals from multiple voxels increases the sensitivity for stimulus information that may be represented in patterns of responses, rather than in individual voxels (*Santoro et al., 2017*).

## Broadened temporal tuning in aging auditory cortex

Under the assumption that the reconstruction accuracy reflects tuning properties of neuronal populations, the observed decrease in selectivity reveals a broadening of temporal tuning functions with age. This finding provides mechanistic insights into the changes of temporal processing in the aging auditory cortex: The broadened temporal tuning, reminiscent of an age-related decrease in temporal precision in the midbrain (*Anderson et al., 2012*), may underlie the difficulties older adults experience in speech comprehension in noisy environments (*Anderson et al., 2011*). While the present study lacks more direct measures of encoding along the auditory pathway, it is important to note that rate selectivity in the older listeners was statistically unrelated to their pure-tone audiograms

(this holds true for both presented measures of tuning, the selectivity index [Figure 5] and the variance across temporal rates [*Figure 6*]). Thus, the broadening of temporal tuning is not explicable merely by peripheral hearing loss. Future studies need to carefully assess the consequences of altered cortical temporal tuning on behavioural and psychoacoustic performance (*Erb et al., 2019b*; *Flinker et al., 2019*; *Holmes and Griffiths, 2019*), a measure that was precluded by the use of natural sounds in the current study.

The observation of decreased temporal selectivity may seem at odds with previous accounts of elevated cortical responses to the temporal speech envelope in aging. Evidence for better reconstruction of temporal envelopes from older listeners' MEG signals (*Presacco et al., 2016*) and larger N1 amplitudes in older adults with widened auditory filters (*Herrmann et al., 2016*; *Henry et al., 2017*) suggest an enhanced cortical envelope representation. However, an inflated cortical representation of the speech envelope has been argued to rather be a maladaptive response as compensation of the lack of temporal precision in the midbrain (*Anderson et al., 2012*).

Neuronal coding of temporal modulations follows two distinct principles, that is rate and temporal coding (*Joris et al., 2004*). While rate coding constitutes a variation of overall spike rate as a function of modulation frequency, temporal coding is achieved through phase-locking to the stimulus envelope. Fast temporal modulations (>50 Hz) are typically encoded through the rate code, while neurons tuned to slow temporal rates (<50 Hz) mostly synchronize to the sounds' modulations (temporal code, *Sachs, 1984*; *Joris et al., 2004*). It remains unclear how hemodynamic responses reflect these two types of neuronal coding and how those are in turn related to the observed age-related broadening of tuning functions. Evidence from electrophysiology suggests a change in coding strategy across neuronal populations in A1 from less temporal coding towards more rate coding of temporal modulations in aged animals (*Ng and Recanzone, 2018*).

Interestingly, the observed age-related decrease in selectivity to temporal modulations was restricted to areas HG/HS and PT, the putative homologues of areas A1 and CL in the macaque (*Kaas and Hackett, 2000*). Both areas have been shown to exhibit altered temporal encoding in older macaques (*Ng and Recanzone, 2018*). In a direct comparison of the vector strengths (a metric of periodicity of the neuronal response to a modulated signal) as a function of the inter-stimulus interval, responses differed in young A1 and CL neurons such that A1 neurons had higher vector strength. This is consistent with the notion that CL neurons process predominantly spatial (rather than temporal) information. Therefore, so the theory, CL neurons would not necessarily need to pass on the temporal coding of A1 neurons. This difference between belt and core areas was abandoned in the older animals, where aged A1 were similar to both the aged and young CL neurons, indicating that the temporal fidelity of A1 responses decreases with age.

However, a good temporal fidelity of A1 neurons is thought to be critical for temporal processing. We speculate that such an age-related decline in temporal processing may be linked to an imbalance of excitation and inhibition observed in aging (*Voytek et al., 2015*). Inhibitory (e.g. GABAergic or glycinergic) neurotransmitters have been shown to increase response synchrony to modulated stimuli in both the cochlear nucleus and the inferior colliculus (*Koch and Grothe, 1998*; *Backoff et al., 1999*, for review see *Caspary et al., 2008*), suggesting that inhibitory neurotransmission is of particular importance to precise neural timing and, thus, the adequate tracking of temporal sound features.

An inherent challenge to age group comparisons is to distinguish any unspecific decrease of SNR in the data of older participants (e.g., due to vascular changes that impact the BOLD response *Garrett et al., 2017* or due to the typical increase in movement artefacts) from specific age-related sensory processing changes. Most reassuringly, however, a multiple regression model taking into account overall goodness of the encoding model fit (i.e., sound identification accuracy) did not alter the observed relationship between age and temporal rate tuning.

## Conclusions

The present study aimed to narrow the gap between recent progress in modelling the neural processing of acoustic features from natural stimuli on the one hand, and the understanding of potential senescent changes in these cortical stimulus representations on the other hand. Although the large-scale topographic organization of acoustic features appears preserved in the auditory cortex of older compared to younger listeners, age-related differences in the marginal profiles of multi-voxel MTFs were evident. Tuning to slow temporal rates which abounds in natural sounds and especially in

speech (*Erb et al., 2019a*) was markedly sharper in young compared to older participants. Consistent with previous findings in the macaque, these results suggest that temporal rate selectivity in auditory cortex declines in normal aging. The specificity of this decline, confined to primary auditory and adjacent areas and sparing tonotopic representations, opens a new lead in the ongoing search for tractable neurobiological signatures of older adults' widely observed deficits in speech comprehension in noisy environments.

## Materials and methods

### Participants

We invited a total of *n = 75* participants for scanning from which we had to exclude one participant due to excessive movement, three participants due to incidental neurological findings, five participants due to their inability to understand speech in noise, one participant due to broken headphones, one participant due to age (43 years, i.e., could not be assigned to young or older group). The remaining *n = 64* participants were right-handed, young (*n = 33*; aged 18–32, mean 24.7 years18 female) and older (*n = 31*; aged 51–78, mean 63.8 years, 15 female) native German speakers. Simulations (custom matlab code) showed that a two-sample permutation test at a conventional type I error rate of 5% with a sample size of *N = 27* per group can detect medium to large effects of Cohen's *d* = 0.75 (as can be expected for the neural fMRI measures under consideration, for example *Alavash et al., 2019*) with a satisfactory power of 80%. The power of our procedure reduces accordingly if the true effect in the population is smaller but remains over 60% also for true effects closer to Cohen's *d* = 0.5.

Younger participants had self-reported normal hearing. Older participants' hearing ranged from normal hearing to mild hearing loss. The older participants were part of a cohort which is regularly tested in the lab (e.g., *Alavash et al., 2019*). They were recruited based on their audiograms that had been acquired within the two years prior to the experiment (for audiograms see *Figure 1—figure supplement 1a*) and were excluded from the study if the pure-tone-average (PTA) of the better ear exceeded 30 dB HL. On average, hearing thresholds are expected to increase by approximately 0.25–1.7 dB HL per year at the age range of 48–79 years (*Wiley et al., 2008*). Thus, we considered 2-year-old audiograms to be valid. All participants gave informed consent and were financially compensated or received course credit. All procedures were approved by the local ethics committee of the University of Lübeck (ethical approval AZ 16–107).

### Stimuli and task

Participants listened to 64 min of a freely narrated audiobook (Hertha Müller, '*Die Nacht ist aus Tinte gemacht*') presented against a competing stream of resynthesized natural sounds ('sound textures'; *McDermott and Simoncelli, 2011*) at 0 dB SNR. Textures were synthesized from the spectro-temporal modulation content of a large set of real-life sounds (*n = 192*), including speech and vocal samples, music pieces, animal cries, scenes from nature and tool sounds that had been used in previous studies (*Moerel et al., 2013*; *Santoro et al., 2017*). Texture synthesis parameters were as follows: Frequency range = 0.02–10 kHz, number of frequency bands = 30, sampling rate = 20 kHz, temporal modulation range = 0.5–200 Hz, sampling rate = 400 Hz; maximum number of iterations = 60. Texture exemplars of 5 s length were concatenated to form the background stream. The order of exemplars was pseudo-randomized across participants (in four different sound orders). In total, each exemplar was repeated four times.

Participants were asked to listen to the story and answer three four-choice questions on its semantic content after each run (24 questions in total); chance level was thus proportion correct = 0.25. Four-answer choice questions were displayed on a screen and participants responded via a button box in their right hand. Both the young and older group performed on average above chance (see *Figure 1—figure supplement 1b*), but young participants performed significantly better than older participants (mean proportion correct difference = 0.1, p=0.009 [permutation test], Cohen's *d* = 0.65).

## Acquisition of MRI data

We acquired functional and structural MRI at 3T (Siemens Magnetom Skyra) with a 64-channel RF head array coil.

Functional T2*-weighted data were acquired using an echo planar imaging sequence. We collected eight runs of eight minutes each (each run contained 519 volumes) using continuous scanning. The acquisition parameters were as follows: repetition time (TR) = acquisition time (TA) = 947 ms, echo time (TE) = 28 ms, acceleration factor = 4, flip angle = 60°, field of view (FOV) = 200×200 mm, 52 slices; voxel size = 2.5 mm isotropic (whole-brain coverage). Field maps for intensity inhomogeneity correction were acquired after every second run (TR = 610 ms, TE1 = 4.92, TE2 = 7.38, flip angle = 60°, FOV = 200×200 mm, 62 slices, voxel size = 2.5 mm isotropic).

Anatomical T1-weighted images were acquired using an MPRAGE sequence (TR = 2400 ms, time to inversion [TI]=1000 ms, TE = 3.16 ms, flip angle = 8°, FOV = 256×256 mm, number of slices = 176, voxel size = 1 mm isotropic, GRAPPA acceleration factor = 2). T2-weighted images were collected at the end of the session (TR = 3200 ms, TE = 449 ms, FOV = 256×256 mm, number of slices = 176, voxel size = 1 mm isotropic, GRAPPA acceleration factor = 2).

## Preprocessing of MR data

Results included in this manuscript are based on a preprocessing pipeline of *fMRIPprep* 1.2.4 which is based on *Nipype* 1.1.6 (*Esteban et al., 2019*). The following two paragraphs on preprocessing are based on an automatically generated output of *fMRIPprep*.

### Anatomical data preprocessing

The T1-weighted (T1w) image was corrected for intensity non-uniformity (INU) using N4BiasFieldCorrection (ANTs 2.2.0) and used as T1w-reference throughout the workflow. The T1w-reference was then skull-stripped using antsBrainExtraction.sh (ANTs 2.2.0), using OASIS as target template. Brain surfaces were reconstructed using recon-all (FreeSurfer 6.0.1), and the brain mask estimated previously was refined with a custom variation of the method to reconcile ANTs-derived and FreeSurfer-derived segmentations of the cortical gray-matter of Mindboggle. Spatial normalization to the ICBM 152 Nonlinear Asymmetrical template version 2009c was performed through nonlinear registration with antsRegistration (ANTs 2.2.0), using brain-extracted versions of both T1w volume and template. Brain tissue segmentation of cerebrospinal fluid (CSF), white-matter (WM) and gray-matter (GM) was performed on the brain-extracted T1w using fast (FSL 5.0.9).

### Functional data preprocessing

For each of the eight BOLD runs, the following preprocessing was performed. First, a reference volume and its skull-stripped version were generated using a custom methodology of *fMRIPrep*. A deformation field to correct for susceptibility distortions was estimated based on a field map that was co-registered to the BOLD reference, using a custom workflow of *fMRIPrep*. Based on the estimated susceptibility distortion, an unwarped BOLD reference was calculated for a more accurate co-registration with the anatomical reference. The BOLD reference was then co-registered to the T1w reference using bbregister (FreeSurfer) which implements boundary-based registration. Co-registration was configured with nine degrees of freedom to account for distortions remaining in the BOLD reference. Head-motion parameters with respect to the BOLD reference (transformation matrices, and six corresponding rotation and translation parameters) are estimated before any spatiotemporal filtering using mcflirt (FSL 5.0.9). BOLD runs were slice-time corrected using 3dTshift from AFNI 20160207. The BOLD time-series (including slice-timing correction when applied) were resampled onto their original, native space by applying a single, composite transform to correct for head-motion and susceptibility distortions. In a control analysis, ICA-based Automatic Removal Of Motion Artifacts (AROMA) was used to generate a variant of data that is non-aggressively denoised (*Pruim et al., 2015*). All analyses (see below) were performed in native space. Subsequently, results (best feature maps) were resampled to surfaces on the *fsaverage5* template using mri_vol2surf (FreeSurfer).

## Modulation representation

The modulation content of the stimuli was obtained by filtering the sounds within a biologically plausible model of auditory processing (*Chi et al., 2005*). This auditory model consists of an early stage that models the transformations that acoustic signals undergo from the cochlea to the midbrain; and a cortical stage that accounts for the processing of the sounds at the level of the auditory cortex. We derived the spectrogram and its modulation content using the 'NSL Tools' package (available at http://www.isr.umd.edu/Labs/NSL/Software.htm) and customized Matlab code (The MathWorks Inc, Matlab 2014b/2018a).

Spectrograms for all sounds were obtained using a bank of 128 overlapping bandpass filters with equal width ($Q_{10dB}$ = 3), spaced along a logarithmic frequency axis over a range of $f$ = 180–7040 Hz. The output of the filter bank was band-pass filtered (hair cell stage). A midbrain stage modelled the enhancement of frequency selectivity as a first-order derivative with respect to the frequency axis, followed by a half-wave rectification and a short-term temporal integration (time constant $\tau$ = 8 ms).

Next, the auditory spectrogram was further analyzed by the cortical stage, where the modulation content of the auditory spectrogram was computed through a bank of 2-dimensional filters selective for a combination of spectral and temporal modulations. The filter bank performs a complex wavelet decomposition of the auditory spectrogram. The magnitude of such decomposition yields a phase-invariant measure of modulation content. The modulation selective filters have joint selectivity for spectral and temporal modulations, and are directional, that is they respond either to upward or downward frequency sweeps.

Filters were tuned to spectral modulation frequencies of $\Omega$ = [0.3, 0.4, 0.8, 1.3, 2.3, 4] cyc/oct, temporal modulation frequencies of $\omega$ = [1, 2, 4, 8, 16, 32] Hz, and centre frequencies of $f$ = [232, 367, 580, 918, 1452, 2297, 3633, 5746] Hz. Our rationale for this choice of values was to use a decomposition roughly covering the temporal and spectral modulations present in the acoustic energy of natural sounds we used (for spectro-temporal modulation content of the sounds see *Figure 1—figure supplement 2*). To avoid overfitting, for the decoding analyses in regions of interest (ROIs), we reduced the number of features such that filters were tuned to spectral modulation frequencies of $\Omega$ = [0.25, 0.5, 1, 2, 4] cyc/oct, temporal modulation frequencies of $\omega$ = [1, 2.4, 5.7, 13.5, 32] Hz, and centre frequencies of $f$ = [277, 576, 1201, 2502, 5213] Hz.

The filter bank output was computed at each frequency along the tonotopic axis and then averaged over time. This resulted for the encoding (ROI decoding) analysis in a representation with 6 (5) spectral modulation frequencies × 6 (5) temporal modulation frequencies × 8 (5) tonotopic frequencies = 288 (125) parameters to learn. The time-averaged output of the filter bank was averaged across the upward and downward filter directions (*Santoro et al., 2014*). Those processing steps were applied to all stimuli, resulting in an [$N$ x $F$] feature matrix $S$ of modulation energy, where $N$ is the number of sounds, and $F$ is the number of features in the modulation representation.

Prior to acoustic feature extraction, the continuous sounds (~8 min per run) were cut into snippets of the length of the TR (947 ms) resulting in a total of $n$ = 4152 sounds which were subdivided into training ($n$ = 3114) and test sounds ($n$ = 1038) for the cross-validation procedure (see below). This resulted in an equivalent temporal resolution of the feature matrix and the fMRI data.

Note that for the encoding analysis (see below), the number of parameters to estimate is thus smaller than the number of observations in the training set ($n$ = 3114 training sounds). In the decoding analysis, the number of fitted features is limited by the number of voxels instead (*Equation 5*). Each feature was convolved with the standard double gamma model for the hemodynamic response function peaking at 4 s.

## Encoding and decoding models

We applied two modelling approaches to fMRI data as described in *Erb et al., 2019b*; *Santoro et al., 2014*; *Santoro et al., 2017* (*Figure 1*). In a first *univariate encoding* approach, we calculated a modulation transfer function (MTF) for each individual voxel. The MTF characterizes how faithfully the modulation content of the stimulus gets transferred to the voxel. By assigning the feature value with the maximal response to each voxel, we obtained maps of a voxel's best features across the auditory cortex.

In a second *multivariate decoding* approach, data from voxels were jointly modelled within a model-based decoding framework. The combined analysis of signals from multiple voxels increases

the sensitivity for stimulus information that is represented in patterns of activity, rather than in individual voxels. Further, the accuracy with which those features can be reconstructed provides an explicit measure of the amount of information about sound features available in cortex.

## Univariate encoding analysis: Model estimation

Based on the training data only, the fMRI activity $Y_i$ [$N_{train} \times 1$] at voxel $i$ was modeled as a linear transformation of the feature matrix $S_{train}$ [$N_{train} \times F$] plus a noise term $n$ [$N_{train} \times 1$]:

$$Y_i = S_{train}C_i + n \tag{1}$$

where $N_{train}$ is the number of sounds in the training set, and $C_i$ is an [$F \times 1$] vector of model parameters, whose elements $c_{ij}$ quantify the contribution of feature $j$ to the overall response of voxel $i$. Columns of matrices $S_{train}$ and $Y_i$ were converted to standardized z-scores. Therefore, *Equation 1* does not include a constant term. The solution to *Equation 1* was computed using kernel ridge regression (*Hoerl and Kennard, 1970*). The regularization parameter $\lambda$ was selected independently for each voxel via generalized cross-validation (*Golub et al., 1979*). The search grid included 25 values between $10^{-6}$ and $10^{6}$ logarithmically spaced with a grid grain of $10^{0.5}$.

To obtain more stable estimates of the voxels' feature profiles, this computation was performed in a fourfold cross-validation procedure within each participant using different subsets of the eight runs. For each iteration, two out of the eight runs were selected and left out for testing, resulting in a subset of 3114 training sounds on which the estimation was performed and 1038 test sounds. In this way, we obtained four estimates of each voxel's feature profile which were averaged across iterations.

## Model evaluation

To evaluate the model's prediction accuracy, we performed a sound identification analysis (*Kay et al., 2008*). To this end, we used the fMRI activity patterns predicted by the model to identify which of the test sounds had been heard. Given the trained model $\tilde{C}$ [$F \times V$], and the feature matrix $S_{test}$ [$N_{test} \times F$] for the test set, the predicted fMRI activity $\hat{Y}_{\text{test}}$ [$N_{test} \times V$] for the test sounds was obtained as:

$$\hat{Y}_{test} = S_{test}\,\tilde{C} \tag{2}$$

Then, we computed for each stimulus $s_k$ the correlation $r_s$ between its predicted fMRI activity $\hat{Y}_{\text{test}}(s)$ [$1 \times V$] and all measured fMRI responses $Y_{test}(s)$ [$1 \times V$]. The rank of the correlation between predicted and observed activity for stimulus $s_k$ was used as a measure of the model's ability to correctly match $Y_{\text{test}}(s)$ with its prediction $\hat{Y}_{\text{test}}(s)$. The rank was then normalized between 0 and 1 as follows to obtain the sound identification score $m$ for stimulus $s$ (*Santoro et al., 2014*):

$$m_s = 1 - \frac{rank(r_s) - 1}{N_{test} - 1} \tag{3}$$

Note that $m = 1$ indicates a correct match; $m = 0$ indicates that the predicted activity pattern for stimulus $s_i$ was least similar to the measured one among all stimuli. Normalized ranks (sound identification scores) were computed for all stimuli in the test set, and the overall model's accuracy was obtained as the mean of the sound identification scores across stimuli.

## Topographical best-feature maps

The response profiles for temporal modulation, spectral modulation and frequency were computed as marginal sums of the estimated stimulus-activity mapping function $C$ of the frequency-specific modulation model by summing across irrelevant dimensions. For example, to obtain the temporal modulation transfer function (tMTF), we summed across the spectral modulation and frequency dimension:

$$tMTF(\omega) = \sum_{\Omega}\sum_{f} C(\omega, \Omega, f) \tag{4}$$

To calculate profiles for the spectral modulation transfer function (sMTF) and frequency transfer

function (fTF), we correspondingly summed across irrelevant dimensions. The voxels' best features were defined as the maximum of the tMTF, sMTF and fTF, respectively. Cortical maps were generated by colour-coding the voxels' preferred values and projecting them onto an inflated representation of the cortex. To obtain group maps, individual maps were transformed to FreeSurfer's *fsaverage5* space and averaged.

## Multivariate decoding analysis: Model estimation

In the multivariate decoding analysis, we evaluated the fidelity with which regions of interest (ROIs) in auditory cortex encode acoustic features by estimating decoders. In addition to a region including the whole auditory cortex, we selected five ROIs based on anatomical criteria (using the FreeSurfer labels): Heschl's gyrus/sulcus (HG/HS), planum temporale (PT), planum polare (PP), superior temporal gyrus (STG) and sulcus (STS). Additionally, we selected three control ROIs, namely the visual cortex (V1, calcarine sulcus), middle frontal (MFG) and superior parietal gyrus (SPG). For each individual in each ROI, a linear decoder was trained for every feature of the modulation space based on the training data only (*Santoro et al., 2017*). Each stimulus feature $S_j$ [$N_{train} \times 1$] was modelled as a linear transformation of the multi-voxel response pattern $Y_{train}$ [$N_{train} \times V$] plus a bias term $b_j$ and a noise term $n$ [$N_{train} \times 1$] as follows:

$$S_j = Y_{train}C_j + b_j 1 + n \tag{5}$$

where $N_{train}$ is the number of sounds in the training set, V is the number of voxels, one is a [$N_{train} \times 1$] vector of ones, and $C_j$ is a [$V \times 1$] vector of model parameters, whose elements $c_{ji}$ quantify the contribution of voxel $i$ to the encoding of feature $j$. All parameters in *Equation 5* were estimated within the same ridge-regression cross-validation scheme as *Equation 1*. Here, the regularization parameter λ was determined independently for each feature by generalized cross validation using the identical search grid as in the univariate encoding model (see above).

## Estimation of multi-voxel modulation transfer functions (MTFs)

Decoders were estimated on the train runs (see above) and tested on the test runs. Given the trained model $\tilde{C}$ [$V \times F$], and the patterns of fMRI activity for the test sounds $Y_{test}$ [$N_{test} \times V$], the predicted feature matrix activity $S_{test}$ [$N_{test} \times F$] for the test sounds was calculated as:

$$\hat{S}_{test} = Y_{test}\tilde{C} \tag{6}$$

Predictions for all features from the test sets were concatenated and decoders were assessed individually by computing the Pearson's correlation coefficient (*r*) between a predicted and a given actual stimulus feature. For decoding from the whole auditory cortex (ROIs), this resulted in 288 (125) correlation coefficients, which represented the MTF. To obtain marginal profiles of the MTFs, we averaged across the momentarily irrelevant dimensions, respectively.

Statistical significance of the reconstruction accuracy was assessed with permutation testing. For each stimulus feature we computed the null-distribution of accuracies at the single-subject level by randomly permuting (1,000 times) the stimulus labels of the test sounds and computing the correlation coefficient with the predicted features for each permutation. Note that to reduce computing time and resources, we limited the number of permutations to 1,000 whenever encoding and decoding models had to be estimated for each permutation at the single-subject level. For less computationally intense calculations (group comparisons), we used the default of $n = 10,000$ permutations.

In order to preserve the temporal correlations among sounds, the predictors were phase scrambled by using the fast Fourier transform (fft), randomly permuting the phase angles of the fft and calculating the inverse fft before convolution with the hemodynamic response function. The empirical chance level of correlation $r_{chance}$ was defined as the mean of the null distribution. The *p*-value was computed as the proportion of permutations that yielded a correlation equal to or more extreme than the empirical one. The reconstruction accuracies were z-scored relative to the empirical null distribution.

## Post-hoc statistical analysis of marginal MTFs

Group marginal profiles of MTFs were obtained as the mean of all individual marginal MTFs. To assess the statistical significance of the observations on the MTF's marginal profiles, we performed the following post-hoc analyses.

## Selectivity index

Based on the marginal profiles of the MTFs, we quantified how selectively a brain region is tuned towards processing the acoustic feature with the highest reconstruction accuracy relative to all other features using a selectivity index

$$SI = \frac{\max_{j\in[k]}(r_j) - \operatorname{mean}_{j\neq j_{max}}(r_j)}{\max_{j\in[k]}(r_j) + \operatorname{mean}_{j\neq j_{max}}(r_j)} \tag{7}$$

where $r_j$ is the reconstruction accuracy at feature $j$, $k$ is the number of features, $j_{max}$ is the index of the maximal reconstruction accuracy. $SI$ was calculated for each acoustic dimension separately (i.e., rate, scale or frequency) and compared between age groups using an exact permutation test by calculating the mean group difference and permuting the age group labels 10,000 times.

## Acknowledgements

This research was funded by an ERC consolidator Grant (ERC-CoG-2014–646696 'AUDADAPT' to JO) and the German Research Foundation (DFG; OB 352/2–1). Martin Göttlich helped with MR sequences. Anne Herrmann, Malte Naujokat, Clara Mergner, and Anne Ruhe helped acquire the data. We are grateful for the methods and analysis tools developed at the Department for Cognitive Neuroscience, Maastricht University, in particular by Federico De Martino, Roberta Santoro and Elia Formisano which were central for the current project. We thank the members of the Auditory Cognition group as well as the editors and reviewers for very constructive feedback on the present study.

## Additional information

### Competing interests

Jonas Obleser: Reviewing editor, *eLife*. The other authors declare that no competing interests exist.

### Funding

| Funder | Grant reference number | Author |
| --- | --- | --- |
| H2020 European Research Council | ERC-CoG-2014-646696 "AUDADAPT" | Jonas Obleser |
| Deutsche Forschungsgemeinschaft | OB 352/2-1 | Jonas Obleser |

The funders had no role in study design, data collection and interpretation, or the decision to submit the work for publication.

### Author contributions

Julia Erb, Conceptualization, Resources, Data curation, Software, Formal analysis, Supervision, Validation, Investigation, Visualization, Methodology, Writing - original draft, Project administration, Writing - review and editing; Lea-Maria Schmitt, Conceptualization, Resources, Data curation, Software, Investigation, Methodology, Project administration, Writing - review and editing; Jonas Obleser, Conceptualization, Formal analysis, Supervision, Funding acquisition, Validation, Methodology, Writing - original draft, Writing - review and editing

## Author ORCIDs

Julia Erb (iD) https://orcid.org/0000-0002-3440-7269
Lea-Maria Schmitt (iD) http://orcid.org/0000-0002-9356-2234
Jonas Obleser (iD) http://orcid.org/0000-0002-7619-0459

## Ethics

Human subjects: All participants gave informed consent and were financially compensated or received course credit. All procedures were approved by the local ethics committee of the University of Lübeck (ethical approval AZ 16-107).

## Decision letter and Author response

Decision letter https://doi.org/10.7554/eLife.55300.sa1
Author response https://doi.org/10.7554/eLife.55300.sa2

# Additional files

## Supplementary files

• Transparent reporting form

## Data availability

MRI data and custom code to reproduce all essential findings are publicly available on the Open Science Framework (OSF).

The following dataset was generated:

| Author(s) | Year | Dataset title | Dataset URL | Database and Identifier |
|---|---|---|---|---|
| Erb J, Schmitt L-M, Obleser J | 2020 | GRASP | https://osf.io/zbuah/ | Open Science Framework , 10.17605/OSF.IO/28R57 |

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
