## [Decision Letter]

**Acceptance summary:**

Using fMRI and computation modeling, this study explores how age affects cortical responses to natural sounds with different spectrotemporal properties. Using both encoding and decoding analyses, the authors demonstrate that older listeners have broadened temporal rate tuning compared to younger listeners, but show no difference in spectral tuning.

**Decision letter after peer review:**

Thank you for submitting your article "Temporal selectivity declines in the aging human auditory cortex" for consideration by *eLife*. Your article has been reviewed by Barbara Shinn-Cunningham as the Senior Editor, Ingrid Johnsrude as Reviewing Editor, and two reviewers. The following individual involved in review of your submission has agreed to reveal their identity: Jonathan Z Simon (Reviewer #1).

The reviewers have discussed the reviews with one another and the Reviewing Editor has drafted this decision to help you prepare a revised submission. In recognition of the fact that revisions may take longer than the time we typically allow, until the research enterprise restarts in full, we will give authors as much time as they need to submit revised manuscripts.

Summary:

The researchers use fMRI and computational modeling to compare auditory cortical responses to natural sounds across subjects of varying ages (cross-sectional design), according to the spectrotemporal properties of the sounds.

Approaching age-based neural dedifferentiation (or neural "tuning") in the auditory domain has not been well handled in the literature to date. In that sense, reviewers found this work to be interesting, particularly with respect to the detailed and nuanced treatment the authors give to the analysis of auditory stimuli (i.e., rates, scales, frequencies).

The reviewers found the study to be well designed.

The analysis proceeds in two steps: first using an encoding framework, and then using a decoding framework. The authors find that older listeners have broadened temporal rate tuning compared to younger listeners, in contrast to spectral tuning which does not show such a difference.

The reviewers commented that the analysis methods are employed well and described clearly. It is evident that there is great attention to detail in the analysis, both in the formalism and the statistics.

The benefits and tradeoffs between the encoding and decoding approaches are handled well, and both reviewers commented on how nice it was to see both frameworks used on the same data.

Essential revisions:

The reviewers raise a number of concerns that must be adequately addressed before the paper can be accepted. Some of the required revisions may require further experimentation within the framework of the presented studies and techniques.

1) Subsection “Univariate encoding analysis: Model estimation”: How were subject data incorporated in the cross-validation scheme? It is clear how sounds were split, but not how subject data were handled. If all subjects are included and you simply split on stimuli, then the data are technically not "independent" per se in the classic between-subjects sense…one can always argue bias is present by having all subjects contribute at each cross-validation stage.

2) Subsection “Multivariate decoding analysis: Model estimation”: It seems to be a major choice to have a single sensory control region (calcarine sulcus) given the task type. Please justify this more.

3) Subsection “Sound identification accuracies”: the joint stimuli (texture + speech) are somehow "uniquely" identified despite texture being repeated 4x. Would results likely be even better without this repetition? Why was texture repetition required at all?

4) Subsection “Decoding results”: It was not clear why decoding accuracy should be highest at these values of freq, scale, and rate. Was there a hypothesis regarding these levels?

5) The age-based result in Figure 5E is questionable. This was plotted as a scatterplot in WebPlotDigitizer to examine the impact of data on the leftmost side of the plot (which visually are pulling the slope negative). Using a Spearman corr reduces the corr value to -.21 (p=.27). Holding out a single high leverage case (only the most extreme value detected by computing Cook's distance) also reduces the Pearson correlation to -.28 (p=.14) and Spearman to -.13 (p=.50). And taking out the three cases identified by a Cook's rule of thumb (Standard Cook's cutoff = 4/n), Pearson and Spearman corrs are reduced to -.19 and -.05 respectively.

6) The uni- and multivariate outliers need to be better addressed in Figure 5B/C as well. The distributions are not terribly well behaved. For example, without what seems like a young adult outlier at the highest SI level in Figure 5C, does the group effect remain?

7) The same should be investigated for Figure 6B also; the right tail of the young group appears to pull up the young adult mean in that case too. It does appear that Figure 6A may hold up however, but this needs to be verified.

8) Because of a lack of clarity in the results as they currently sit, little can be made of the current Discussion section, various aspects of what is discussed may fall away once the data are reanalyzed. The Discussion section (and Abstract) should therefore be revised in light of what the reanalysis shows.

---

## [Author Response]

Summary:The researchers use fMRI and computational modeling to compare auditory cortical responses to natural sounds across subjects of varying ages (cross-sectional design), according to the spectrotemporal properties of the sounds.Approaching age-based neural dedifferentiation (or neural "tuning") in the auditory domain has not been well handled in the literature to date. In that sense, reviewers found this work to be interesting, particularly with respect to the detailed and nuanced treatment the authors give to the analysis of auditory stimuli (i.e., rates, scales, frequencies).The reviewers found the study to be well designed.The analysis proceeds in two steps: first using an encoding framework, and then using a decoding framework. The authors find that older listeners have broadened temporal rate tuning compared to younger listeners, in contrast to spectral tuning which does not show such a difference.The reviewers commented that the analysis methods are employed well and described clearly. It is evident that there is great attention to detail in the analysis, both in the formalism and the statistics.The benefits and tradeoffs between the encoding and decoding approaches are handled well, and both reviewers commented on how nice it was to see both frameworks used on the same data.Essential revisions:The reviewers raise a number of concerns that must be adequately addressed before the paper can be accepted. Some of the required revisions may require further experimentation within the framework of the presented studies and techniques.1) Subsection “Univariate encoding analysis: Model estimation”: How were subject data incorporated in the cross-validation scheme? It is clear how sounds were split, but not how subject data were handled. If all subjects are included and you simply split on stimuli, then the data are technically not "independent" per se in the classic between-subjects sense…one can always argue bias is present by having all subjects contribute at each cross-validation stage.

We thank the reviewers for this comment and would like to clarify the cross-validation scheme. Both encoding and decoding models were run at the single-subject level. Therefore, second-level splits are not feasible for the en-/decoding approach applied here. We now also explicitly state in the manuscript that encoding and decoding were not run on the group level: “… this computation was performed in a fourfold cross-validation procedure within each participant…” (subsection “Multivariate decoding analysis: Model estimation”). For review only, we have added Author response image 1 showing the cross-validation scheme:

**Author response image 1. respfig1:** Cross-validation scheme per subject. In a four-fold cross-validation (CV), the eight fMRI runs of each subject were split into six training and two testing runs, such that each run served once as testing run.Legend.

2) Subsection “Multivariate decoding analysis: Model estimation”: It seems to be a major choice to have a single sensory control region (calcarine sulcus) given the task type. Please justify this more.

We had originally chosen primary visual cortex as a control region because it constitutes another primary sensory area. However, we fully agree with the reviewers that a single sensory control region may not be sufficient. In fact, this comment has now helped us a great deal in substantiating further the specificity of the results to auditory regions HG and PT.

In the revised manuscript we have added two more higher-level control areas that are not primary sensory regions, namely the middle frontal and superior parietal gyrus (Figure 7 and Figure 7—figure supplement 3). Note that for all control regions, reconstruction accuracies were low and no age differences in selectivity index (*SI*) were observed (subsection “Feature reconstruction from auditory subregions”):

“As control regions, we chose primary visual cortex (V1, calcarine sulcus) as another primary sensory area, and two higher-level areas, namely the middle frontal (MFG) and superior parietal gyrus (SPG; Figure 7A,G-I).”

3) Subsection “Sound identification accuracies”: the joint stimuli (texture + speech) are somehow "uniquely" identified despite texture being repeated 4x. Would results likely be even better without this repetition? Why was texture repetition required at all?

We believe that the reviewer questions the necessity of the repetition of sound textures. The main reason for the repetition was that we only had a limited number of textures based on natural sounds (*N* = 192) that we had used in previous experiments (Erb et al., 2019). As recurring sound textures were embedded in non-repetitive speech, each sample of the sound mixture had its unique acoustic properties. Also, the sound mixture was cut in snippets of 947 ms ( = 1 TR) and acoustic features were extracted from these snippets for the sound identification analysis. As the length of snippets was not aligned to the length of sound textures, acoustic features were derived from the different fragments of a recurring sound texture. Therefore, we deem sounds uniquely identifiable.

4) Subsection “Decoding results”: It was not clear why decoding accuracy should be highest at these values of freq, scale, and rate. Was there a hypothesis regarding these levels?

We thank the reviewer for this comment which we take as opportunity to clarify our hypothesis on the decoding results (subsection “Feature reconstruction from auditory cortex”):

“Our main hypothesis was that cortical sensitivity is highest for slow temporal modulations, based on previous observations of human (but not monkey) auditory cortex being most sensitive to the modulations present speech (Santoro et al., 2017, Erb et al., 2019). Decoding yielded highest accuracies at frequencies of 230–580 Hz (mean *r* = 0.47) and spectral scales of 0.25 cyc/oct (mean *r* = 0.47), irrespective of age (Figure 5A,B), indicating that brain responses followed the frequency and spectral modulation content of the stimuli (Figure 1—figure supplement 2A). Conversely, for temporal modulations, reconstruction accuracies peaked at rates of 4–8 Hz (mean r = 0.4) in both age groups (Figure 5A,B). Those peaks were not present in the stimuli (Figure 1—figure supplement 2A) and imply preferred processing of temporal rates in the speech-relevant range (Poeppel and Assaneo, 2020), corroborating our hypothesis (see also Discussion section).”

5) The age-based result in Figure 5E is questionable. This was plotted as a scatterplot in WebPlotDigitizer to examine the impact of data on the leftmost side of the plot (which visually are pulling the slope negative). Using a Spearman corr reduces the corr value to -.21 (p=.27). Holding out a single high leverage case (only the most extreme value detected by computing Cook's distance) also reduces the Pearson correlation to -.28 (p=.14) and Spearman to -.13 (p=.50). And taking out the three cases identified by a Cook's rule of thumb (Standard Cook's cutoff = 4/n), Pearson and Spearman corrs are reduced to -.19 and -.05 respectively.

We thank the reviewer for these additional analyses. We agree with the reviewer that either reporting rank-based (Spearman’s) correlations or identifying and excluding outliers is appropriate here.

Therefore, in the revised version, univariate outliers were identified as being outside of the range mean ± 2 standard deviations (SD). Multivariate outliers were identified based on Cook’s distance. Please also note that for group comparisons we used permutation tests, which are robust against distributional assumptions while being more sensitive than rank-based procedures.

The correlation between the rate selectivity index and age proved indeed not robust to outlier control; we have therefore removed this specific result from the manuscript and Figure 5.

Most important to the overall conclusion, however, note that our second, and arguably less noisy measure of tuning selectivity (i.e., variance of reconstruction accuracy across feature bins) proved robust to outliers (Figure 6 in the revised version).

Note further that in a multiple regression conservatively excluding two multivariate outliers based on Cook’s distance, the prediction of rate variance based on age remains significant (Subsection “Selectivity index”):

“In a control analysis, we ensured that these results cannot be explained by the overall slightly worse model fits in the older group as quantified by sound identification accuracy (as evident in Figure 2): In a multiple regression predicting older participants’ variance in rate from chronological age with covariates PTA and sound identification accuracy, neither PTA (*t(26)* = 0.73, *p* = 0.473, permutation test) nor sound identification accuracy (*t(26)* = 1.02, *p* = 0.312) were significant, while age was significant (*t(26)* = -2.8, *p* = 0.003; *adjusted R^2^* = 0.39). Age remained significant when excluding two multivariate outliers identified based on Cook’s distance (*t*(24) = -2.25, *p* = 0.021).”

6) The uni- and multivariate outliers need to be better addressed in Figure 5B/C as well. The distributions are not terribly well behaved. For example, without what seems like a young adult outlier at the highest SI level in Figure 5C, does the group effect remain?

We thank the reviewer for this comment and have looked at the *SI* age comparison with and without outliers in all respective figures (Figure 5, Figure 5—figure supplement 1, Figure 7). Please note that we used permutation tests for group comparisons which are robust against distributional assumptions. This age group comparison for rate *SI* is not affected by outliers as stated in caption of Figure 5 (see comment # 5 above and Subsection “Selectivity index):

“Note that removing the extreme cases in the young group (outliers were defined as exceeding the grand average by ± 2 SD) for rate *SI* does not alter results qualitatively: It reduces the mean age group difference to a still significant *SI* difference = 0.009, *p* = 0.026.”

7) The same should be investigated for 6B also; the right tail of the young group appears to pull up the young adult mean in that case too. It does appear that 6A may hold up however, but this needs to be verified.

We re-ran the group comparisons for the rate *SI* in HG/HS and PT with and without outliers (Figure 7B,C under comment # 2). Note that the age difference in both ROIs remains significant when excluding outliers (see caption of Figure 7 and subsection “Feature reconstruction from auditory subregions”):

“Removing the extreme cases in the young group exceeding the grand average by ± 2 SD reduces the mean *SI* age difference to 0.012 (*p* = 0.008) in HG/HS, and to 0.012 (*p* = 0.04) in PT.”

8) Because of a lack of clarity in the results as they currently sit, little can be made of the current Discussion section, various aspects of what is discussed may fall away once the data are reanalyzed. The Discussion section (and Abstract) should therefore be revised in light of what the reanalysis shows.

We thank the reviewers for all suggested analyses which – we believe – have substantially strengthened the manuscript. In summary, those analyses have led us to remove the correlation of age and rate SI which did not prove robust to outliers. On the other hand, neither the age group results are nor the correlation of rate variance with age were affected by the suggested analyses. Further, adding two control regions of interest to the decoding analysis has substantiated the result of specificity of age group differences in rate tuning to early auditory areas.

We again thank the reviewers for probing our approach with these comments (#5–#8 above) but feel that the revised, much more thoroughly outlier-controlled analyses justify the conclusion of a noteworthy relationship between chronological age and temporal-rate selectivity.